



# Performance of BDS B1 frequency standard point positioning during the main phase of different classes of geomagnetic storms in China and its surrounding area

Junchen Xue[1], Sreeja Vadakke Veettil[2], Marcio Aquino[2], Xiaogong Hu[1], Lin Quan[3], and Dun Liu[4]

[1]Shanghai Astronomical Observatory, Chinese Academy of Sciences, Shanghai, China
[2]Nottingham Geospatial Institute, University of Nottingham, Nottingham, UK
[3]Beijing Institute of Tracking and Communication Technology, Beijing, China
[4]China Research Institute of Radiowave Propagation, Qingdao, China

**Correspondence:** Junchen Xue (jcxue@shao.ac.cn)

**Abstract.** Geomagnetic storms are one of the space weather events. The radio signals transmitted by modern navigation systems suffer from the effects of storms which can degrade the performance of the whole system. In this study, the performance of BeiDou Navigation Satellite System (BDS) B1 frequency standard point positioning in China and its surrounding area during different classes of storms is investigated for the first time. The analysis of the results revealed that BDS B1 frequency standard point positioning accuracy was deteriorated during the storms. The probability of the extrema in the statistics of positioning errors during strong storms is the largest, followed by moderate and weak storms. The positioning accuracy for storms of a similar class is found not to be at the same level. The root mean square error (RMSE) in position for the different classes of storms could be at least tens of centimeters in the East, North and Up directions.

## 1 Introduction

A geomagnetic storm is defined as a period when the ring current gets intense enough to exceed the key threshold of the distur-bance storm time (Dst) index. Geomagnetic storms are induced by the intense and continuing interplanetary convection electric field and the energy injection to the magnetosphere–ionosphere system (Gonzalez et al., 1994). The enhanced interplanetary convection electric field is motivated by a constant southward interplanetary magnetic field (IMF) (Hori et al., 2005). The solar wind carries the coronal magnetic field into the entire heliosphere, thus forming the IMF (Owens and Forsyth, 2013). Based on the signatures in the magnetic field, a geomagnetic storm can be divided into three phases: initial, main, and recovery. The main phase is the principal characteristic of a geomagnetic storm (Gonzalez et al., 1994; Loewe and Prölss, 1997).

The largest global atmospheric effects can be activated by geomagnetic storms (Lastovicka, 1996). Storms can generate disturbances in the ionosphere, which varies with the location of the region under consideration, local time (LT) of the geo-magnetic storm onset and other parameters (Danilov and Lastovicka, 2001). The Equatorial Ionization Anomaly also responds to the effects of storms (Sreeja et al., 2009). The disturbed condition of the ionosphere during geomagnetic storms is known as an ionospheric storm, which can have great effects on radio propagation dependent applications, especially for GNSS single–





frequency users. These effects can usually be corrected by ionospheric models. For BeiDou Navigation Satellite System (BDS) single–frequency users a Klobuchar–style ionospheric navigation model is applied for the corrections (BDS, 2013).

The performance of GNSS may be influenced by the effects of geomagnetic storms. During storms, carrier phase slips in the

Global Positioning System (GPS) signals may occur, which further results in losses–of–lock (LOL) (Rama Rao et al., 2009). Astafyeva et al. (2014) demonstrated that the density of GPS LOL events during the main phase of storms can increase to 0.25% on L1 band and 3% on L2 band during super storms (Dst$\leq$-250nT), and 0.15% on L1 and 1% on L2 during intense storms (-250nT$<$Dst$\leq$-100nT). Especially, the tracking performance of GPS receivers in the high latitudes was investigated for 2015 St. Patrick's day strong storm. The significant scintillation caused by the storm contributed to LOL on GPS L2 band but had little

influence on the tracking of GPS L1 signal (Jin and Oksavik, 2018). Kinematic GPS positioning could also be degraded during the geomagnetic storm induced ionospheric disturbances. The repeatability of kinematic positioning which were estimated by a two–step approach based on double–difference L3 phase measurements reached 12.8, 8.1, and 26.1 cm in North (N), East (E) and Up (U) directions respectively during the 2003 Halloween storm (Bergeot et al., 2011). The accuracy of Real Time Kinematic (RTK) positioning could also be deteriorated during a strong geomagnetic storm (Jacobsen and Schäfer, 2012),

and even during a weak storm at high latitudes (Andalsvik and Jacobsen, 2014). Furthermore, positioning errors of network RTK and Precise Point Positioning (PPP) techniques increased rapidly during the 2015 St. Patrick's day strong storm period (Jacobsen and Andalsvik, 2016). Investigation was also performed on the effect of moderate and weak geomagnetic storms on the performance of GNSS–SBAS in low latitude African region by using a SBAS emulator to simulate specific EGNOS like messages. The SBAS performance in equatorial African regions showed a non–linear relationship with the geomagnetic

storm indices (Abe et al., 2017). Additionally, GPS instrumental biases, including receiver and satellite biases, are routinely estimated with the dual–frequency geometry–free combination. These computations can also be affected during geomagnetic storms (Zhang et al., 2009).

Even though previous studies have revealed the effects of individual or several geomagnetic storms on the Earth's upper atmosphere and GNSS applications, few papers have studied the performance of BDS based applications during storms, es-

pecially during different classes of geomagnetic storms. As the ionospheric activity could be affected most likely during the geomagnetic storms, GNSS single–frequency users are supposed to be more obviously affected against other positioning modes like PPP during those periods. In this study, the effects of different types of storms on the performance of BDS single frequency standard point positioning, especially for the generally used B1 frequency users, is investigated comprehensively. In addition, the differences in effects between separate storms are studied.

## 2   Methodology

Dst index can be used as a criterion to classify geomagnetic storms (Loewe and Prölss, 1997). In this study, Dst indices were extracted from combined omni files obtained from NASA database (https://omniweb.gsfc.nasa.gov). All storms in solar cycle 24 were analyzed and divided into three classes: Strong, Moderate and Weak. Table 1 states the threshold conditions applied in the classification of storms (see (Gonzalez et al., 1994; Xue et al., 2020)).





**Table 1.** Thresholds implemented in the classification of geomagnetic storms

| Type | Dst (nT) | $\Delta$T(hours) |
| --- | --- | --- |
| Strong | -100 | 3 |
| Moderate | -50 | 2 |
| Weak | -30 | 1 |
| (typical substorm) | | |

The basic strategy for selecting storms is that the Dst should be as minimum as possible and the duration of each storm should be more than 12 hours. To ensure that each storm was independent and not influenced by another storm, a condition was applied that the Dst index for ten days before and after the main phase day must be greater than the minimum value for each individual class of storms. Finally, five cases were chosen for each class of storms from 2015 to 2018 (Xue et al., 2020). The principal characteristic of a geomagnetic storm is its main phase (Loewe and Prölss, 1997). The main phase of storms

including the related Dst values, the start and end epoch, and the duration is presented in Table 2 (see (Xue et al., 2020)). TYPE means three classes of storms. MJD is the modified Julian date. The date with suffix 0 refers to the start epoch while suffix 1 suggests the end epoch. The meaning of suffix for Dst is same with the date. Duration represents the whole period of the main phase.

     BDS observations were collected from the chosen stations on the dates listed in Table 2. Those stations from MGEX network,

namely DAEJ, GMSD, JFNG, LHAZ, HKWS, and HKSL, are distributed in China and its surrounding area. The sampling interval is 30s. The related information, such as geodetic coordinates, receiver and antenna version, is shown in Table 3. The last two columns show the dates when the hardware like receiver or antenna was changed or updated.

     The data were processed in the kinematic mode of standard point positioning (SPP) using BDS single frequency pseudorange observations. Considering the dispersive nature of the ionosphere, only B1 pseudorange was used here. The pseudorange

observation equation is illustrated as follows.

$$B_1 = \rho + dt_r - dt^s + T + I_1 + db_{r1} - db^{s1} + \varepsilon \tag{1}$$

     where, $B_1$ is BDS B1 pseudorange observation, $\rho$ is the geometric range, $dt_r$ is the receiver clock error, $dt^s$ is the satellite clock error, $T$ is the tropospheric delay, $I_1$ is the ionospheric delay, $db_{r1}$ is the receiver differential code bias (DCB), $db^{s1}$ is the satellite DCB, $\varepsilon$ is the noise error.

A conventional option was set for the standard point positioning program. The satellite orbit and clock were computed from IGS navigation data. The tropospheric delays were derived using the Saastamoinen model. The ionospheric delays were calculated by the broadcasted BDS navigation ionospheric model. Time group delays in the broadcast ephemeris were extracted, converted and utilized to compute the satellite DCB. For each epoch, station coordinates and its receiver clock error were





**Table 2.** The main phase of different classes of geomagnetic storms from 2015 to 2018 (STR–Strong, MED–Moderate, MNM–Weak)

| TYPE | MJD0 | YEAR0 | MON0 | DAY0 | DOY0 | HOUR0 | Dst0 (nT) | MJD1 | YEAR1 | MON1 | DAY1 | DOY1 | HOUR1 | Dst1 (nT) | Duration (hours) |
|------|------|-------|------|------|------|-------|-----------|------|-------|------|------|------|-------|-----------|------------------|
| STR | 57098 | 2015 | 3 | 17 | 76 | 5 | 56 | 57098 | 2015 | 3 | 17 | 76 | 22 | -223 | 17 |
| | 57195 | 2015 | 6 | 22 | 173 | 6 | 13 | 57196 | 2015 | 6 | 23 | 174 | 4 | -204 | 22 |
| | 57302 | 2015 | 10 | 7 | 280 | 2 | -9 | 57302 | 2015 | 10 | 7 | 280 | 22 | -124 | 20 |
| | 57375 | 2015 | 12 | 19 | 353 | 22 | 43 | 57376 | 2015 | 12 | 20 | 354 | 22 | -155 | 24 |
| | 58355 | 2018 | 8 | 25 | 237 | 8 | 19 | 58356 | 2018 | 8 | 26 | 238 | 6 | -174 | 22 |
| MED | 57180 | 2015 | 6 | 7 | 158 | 19 | 24 | 57181 | 2015 | 6 | 8 | 159 | 8 | -73 | 13 |
| | 57273 | 2015 | 9 | 8 | 251 | 20 | -2 | 57274 | 2015 | 9 | 9 | 252 | 12 | -98 | 16 |
| | 57406 | 2016 | 1 | 19 | 19 | 19 | 15 | 57407 | 2016 | 1 | 20 | 20 | 16 | -93 | 21 |
| | 57838 | 2017 | 3 | 26 | 85 | 22 | 15 | 57839 | 2017 | 3 | 27 | 86 | 14 | -74 | 16 |
| | 58064 | 2017 | 11 | 7 | 311 | 4 | 25 | 58065 | 2017 | 11 | 8 | 312 | 1 | -74 | 21 |
| MNM | 57544 | 2016 | 6 | 5 | 157 | 8 | 32 | 57545 | 2016 | 6 | 6 | 158 | 6 | -44 | 22 |
| | 57716 | 2016 | 11 | 24 | 329 | 5 | -12 | 57717 | 2016 | 11 | 25 | 330 | 5 | -46 | 24 |
| | 57784 | 2017 | 1 | 31 | 31 | 11 | -5 | 57785 | 2017 | 2 | 1 | 32 | 9 | -45 | 22 |
| | 57920 | 2017 | 6 | 16 | 167 | 7 | 30 | 57920 | 2017 | 6 | 16 | 167 | 23 | -31 | 16 |
| | 58269 | 2018 | 5 | 31 | 151 | 21 | 5 | 58270 | 2018 | 6 | 1 | 152 | 19 | -39 | 22 |

**Table 3.** Longitude, Latitude, Receiver and Antenna version information of stations

| SITEN | LATITUDE | LONGITUDE | RECEIVER | ANTENNA | YEAR | DOY |
|-------|----------|-----------|----------|---------|------|-----|
| DAEJ | 36.40 | 127.37 | TRIMBLE NETR9 | TRM59800.00 | 2017 | 087 |
| | | | | TRM59800.00 | 2015 | 075 |
| GMSD | 30.56 | 131.02 | TRIMBLE NETR9 | TRM41249.00 | 2017 | 311 |
| | | | | TRM59800.00 | 2018 | 151 |
| JFNG | 30.52 | 114.49 | TRIMBLE NETR9 | TRM59800.00 | 2015 | 075 |
| LHAZ | 29.66 | 91.10 | LEICA GR25 | LEIAR25.R4 | 2016 | 157 |
| HKWS | 22.43 | 114.34 | LEICA GR25 | LEIAR25.R4 | 2015 | 353 |
| | | | LEICA GR50 | | 2017 | 031 |
| HKSL | 22.37 | 113.93 | LEICA GR25 | LEIAR25.R4 | 2015 | 353 |
| | | | LEICA GR50 | | 2016 | 329 |





estimated with the Gauss–Newton least square method. The weight was set with the satellite elevation angle. The elevation

mask angle was set to $10°$.

As a result, the station coordinates in cartesian coordinate system were compared with the precise solutions in SINEX files obtained from MGEX products. The related statistics were performed for the main phase period with indices such as minimum (MIN), maximum (MAX), BIAS, and root mean square error (RMSE). The MIN and MAX represent the minimum and maximum of the positioning errors for the three directions (East–E, North–N, Up–U). The BIAS and RMSE are computed

from the positioning errors for each component as well. The formulas are demonstrated as follows.

$$MIN = minimum\{\Delta POS_i\}$$
$$MAX = maximum\{\Delta POS_i\}$$
$$BIAS = <\Delta POS_i>$$
$$RMSE = \sqrt{<\Delta POS_i^2>}$$
$$\Delta POS_i = POS_{ref,i} - POS_{est,i}, i = 1, n \tag{2}$$

Wherein, $<>$ is the average of the variable, $POS_{ref,i}$ is the precise solutions in SINEX files, $POS_{est,i}$ is the solutions obtained in this processing, $n$ is the total number of samples.

## 3  Results and Discussions

The accuracy of BDS B1 frequency SPP solutions during the main phase of different classes of storms is analysed in this section. First, the positioning errors along the three directions for all the stations during a period of 3 days' representative of each class of storm are presented in Fig. 1–Fig. 3. The period shown in each of the figures covers three days before and after the main phase of independent storms. The time range between two vertical red dashed lines is the whole main phase. The left

line indicates the start epoch of the main phase while the right one indicates the end epoch.

As shown in the figures, the positioning errors increased to some extent during the main phase of different classes of storm. The same happened in the recovery phase of the storm as well. For the strong storm shown in Fig. 1, the errors for ENU directions present fluctuations during the main phase of the storm, especially in the U direction in which the errors could be up to approximately 10 meters. In general, the positioning errors could reach about 2 meters in the E direction, about 5

meters in the N direction and about 11 meters in the U direction. However, in this case a larger degradation in positioning errors happened during the recovery phase. This can be attributed to the fact that the Dst values are lower and the geomagnetic disturbance remained intense throughout. It is worth to notice that there are some sharp increases in the positioning errors along the ENU directions of LHAZ station during the main phase of the storm. Nonetheless, there were no similar increases observed for stations HKWS and HKSL which are located at lower latitudes. The increases were observed nearly at 1 LT and

lasted for about 30 minutes, thus indicating that this happened locally and temporarily. The time series of F10.7 cm radio



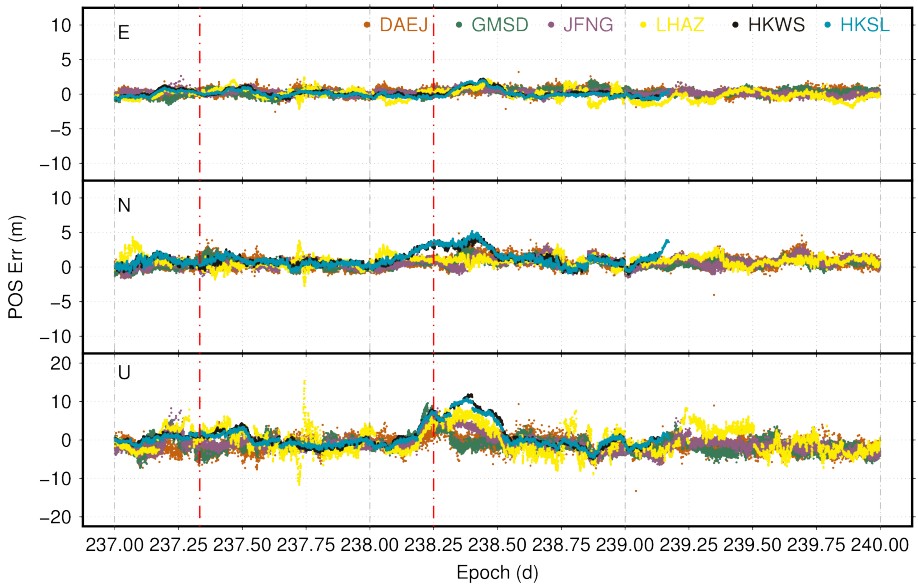

**Figure 1.** Time series of positioning errors for BDS B1 frequency during a strong storm around DOY 238, 2018 (X–axis in GPST, Y–axis in meters)

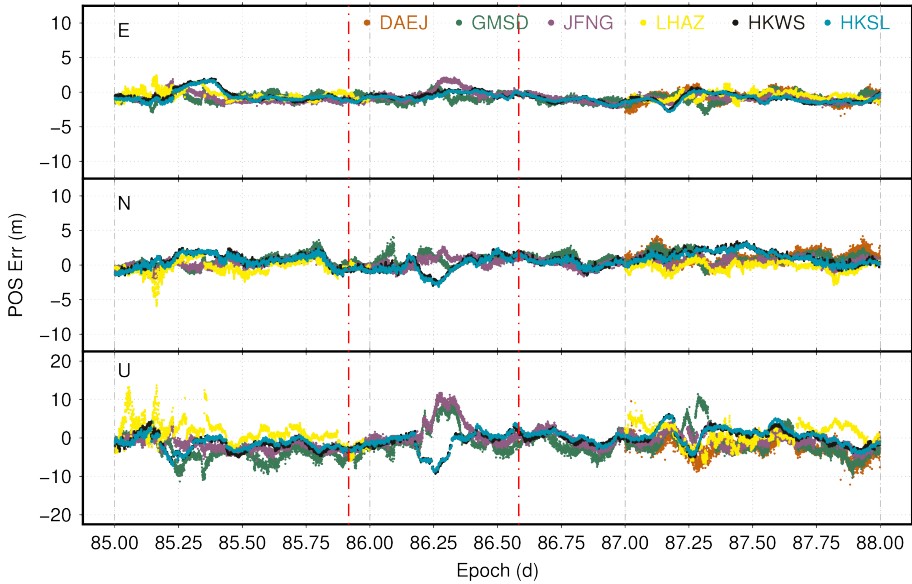

**Figure 2.** Time series of positioning errors for BDS B1 frequency during a moderate storm around DOY 086, 2017 (X–axis in GPST, Y–axis in meters)

flux for the same period was also analysed. however, no sudden variations were observed and the values were under 75 sfu (1 sfu = $10^{-22}w/m^2/hz$), thus implying that there were no abrupt changes in solar activity. Further, BKG Ntrip Client (BNC)



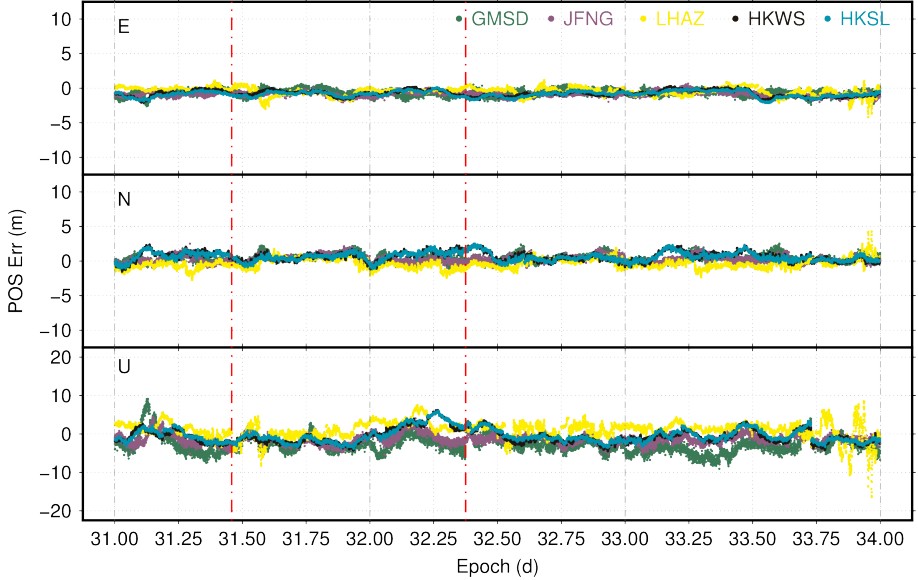

**Figure 3.** Time series of positioning errors for BDS B1 frequency during a weak storm around DOY 032, 2017 (X–axis in GPST, Y–axis in meters)

software (Weber et al., 2016) was used for the data quality check at LHAZ station. The number of satellites and position dilution of precision (PDOP) were computed and shown in Fig. 4. It was found that there were no variations in the number

of satellites during the same period when sharp increases in PDOP values were observed. This indicates that the degradation was directly caused by the poor geometrical structure of constellations. The reason for the short time jumps in PDOP needs further investigation. However, frequent jumps in the number of satellites was found near the end of the whole jump period. The reason for this could be attributed to the receiver issues under the storm. In addition, it can be seen that the change in PDOP values vary and most values were greater than 5 during the whole storm period, thus suggesting that the influence of the

storm on PDOP is strong.

For the moderate storm shown in Fig. 2, the positioning errors of most of the stations varied in all directions during the main phase of the storm. The maxima were about 2 meters in the E direction, 4 meters in the N direction and 10 meters in the U direction. The characteristics of the positioning errors in HKWS and HKSL were different to those of the other stations. This might be attributed to the different version of receiver hardware (LEICA version, see Table 3). There were also changes in the

120 positioning errors along the ENU directions during the recovery phase.

As for the weak storm case shown in Fig. 3, there were slight variations in the positioning errors in the three directions. The positioning errors could reach about 2 meters in the E direction, 2 meters in the N direction and 8 meters in the U direction. The variations in U direction were stronger near the end of the main phase, when the Dst was minimum. Besides this, there were fluctuations in the positioning errors along the U direction during the recovery phase.



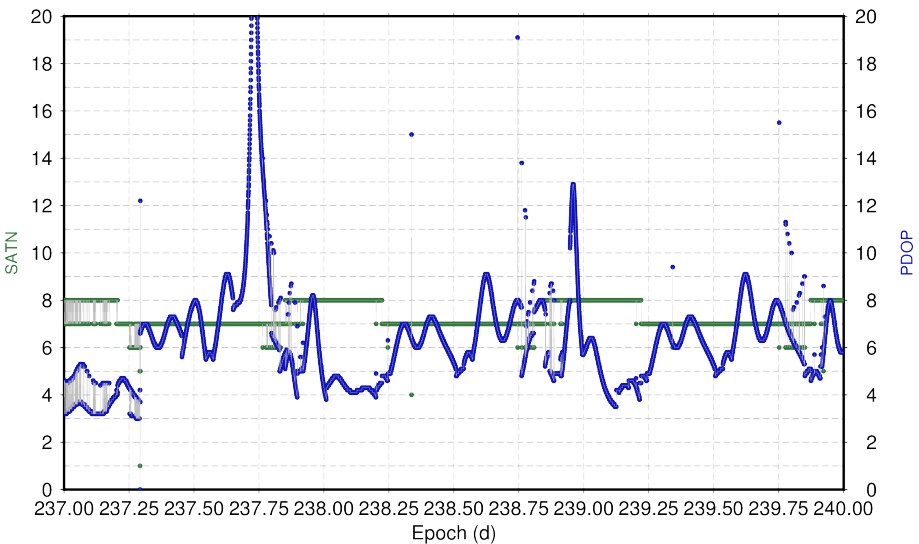

**Figure 4.** The number of satellites and PDOP for LHAZ station during a strong storm around DOY 238, 2018 (X–axis in GPST, left Y–axis for the number of satellites in count, right Y–axis for PDOP in count)

Table 4–Table 6 demonstrate the statistics for BDS B1 frequency positioning errors during the main phase of independent storms. The tables show the indices (MIN, MAX, BIAS, RMSE) for the three different storm classes. The last two rows (MEAN and MEDIAN) are the mean and median of the statistics in each column. Overall, the probability of the extrema in the four statistical indices during strong storms was largest, followed by moderate and weak ones. The difference in positioning errors between different classes of storms was greatest in the U direction.

From Table 4, the maximum positioning error during the strong storms was up to 33 meters, the minimum was nearly -12 meters, BIAS was in the range of 1-4 meters, and the RMSE for ENU directions reached respectively 1.52, 3.25 and 7.92 meters. The MEAN of all RMSEs was 0.95, 1.32, and 3.50 meters in three directions respectively, whilst the MEDIAN was respectively 1.03, 1.12, and 2.96 meters for the ENU directions. Furthermore, the positioning accuracy during different strong storms was not comparable. The accuracy during some storms was better than during others. This indicates not all the strong

storms could have a similar influence on the positioning accuracy. The same feature was observed during the moderate and weak storms as well.

        From Table 5, the maximum of positioning errors during moderate storms could be 12 meters, the minimum was close to -11 meters, BIAS was in the range of 1-3 meters, and RMSE could be up to 1.83, 1.87, and 5.40 meters respectively in the ENU directions. The MEAN and MEDIAN for RMSEs along the three directions were 0.94, 1.09, 3.21 meters and 0.90, 1.07,

3.06 meters respectively.

        From Table 6, the positioning errors during weak storms were generally less than those observed during moderate and strong ones. It can be noticed from Table 6 that LHAZ, HKWS and HKSL stations present an irregular behaviour on MJD 57717 (DOY 329, 2016). Among the three stations, the MAX reached 50.08 meters in the N direction of HKSL, while the



**Table 4.** Statistical indices for BDS B1 frequency positioning errors during strong storms (units: m)

| MJD | SITEN | MIN | | | MAX | | | BIAS | | | RMSE | | |
|---|---|---|---|---|---|---|---|---|---|---|---|---|---|
| 57098 | gmsd | -4.95 | -2.62 | -3.75 | 0.66 | 3.01 | 16.84 | -1.08 | -0.21 | 1.68 | 1.45 | 0.88 | 3.37 |
| | jfng | -3.69 | -3.53 | -4.04 | 1.39 | 4.45 | 10.21 | -1.20 | -0.19 | 1.75 | 1.41 | 1.33 | 2.97 |
| 57196 | gmsd | -2.91 | -2.34 | -9.22 | 0.31 | 1.81 | 7.59 | -1.08 | 0.07 | -0.29 | 1.23 | 0.81 | 3.59 |
| | jfng | -2.26 | -3.51 | -5.23 | 1.10 | 1.90 | 8.03 | -0.52 | -0.19 | 1.87 | 0.91 | 0.97 | 3.26 |
| 57302 | gmsd | -4.71 | -5.17 | -12.09 | 0.69 | 7.01 | 17.00 | -1.32 | -0.12 | -1.08 | 1.52 | 2.26 | 5.29 |
| | jfng | -2.52 | -3.95 | -6.42 | 1.28 | 4.29 | 10.31 | -0.88 | -0.16 | 0.01 | 1.14 | 1.60 | 2.85 |
| 57376 | gmsd | -4.18 | -2.90 | -7.59 | 1.20 | 31.76 | 33.54 | -0.93 | 1.11 | 2.89 | 1.34 | 3.25 | 6.45 |
| | jfng | -2.50 | -2.54 | -4.61 | 3.66 | 4.49 | 22.24 | -0.20 | 0.93 | 4.69 | 1.46 | 1.79 | 7.92 |
| 58356 | daej | -2.54 | -1.31 | -7.65 | 2.15 | 3.94 | 4.87 | 0.25 | 0.43 | -0.98 | 0.54 | 0.80 | 1.98 |
| | gmsd | -1.37 | -0.97 | -5.89 | 1.17 | 2.83 | 9.57 | 0.09 | 0.39 | -0.66 | 0.40 | 0.71 | 2.34 |
| | jfng | -1.07 | -1.33 | -4.17 | 1.43 | 2.06 | 7.54 | 0.17 | 0.38 | -0.98 | 0.41 | 0.61 | 1.83 |
| | lhaz | -2.12 | -2.74 | -11.67 | 2.44 | 3.83 | 15.37 | 0.10 | 0.63 | -0.40 | 0.71 | 1.02 | 2.95 |
| | hkws | -0.77 | -0.51 | -3.30 | 1.33 | 3.60 | 7.93 | 0.09 | 0.95 | 0.23 | 0.41 | 1.22 | 2.20 |
| | hksl | -0.79 | -0.71 | -2.73 | 1.03 | 3.77 | 7.47 | -0.01 | 0.93 | 0.10 | 0.37 | 1.25 | 1.93 |
| MEAN | | -2.60 | -2.44 | -6.31 | 1.42 | 5.63 | 12.75 | -0.47 | 0.35 | 0.63 | 0.95 | 1.32 | 3.50 |
| MEDIAN | | -2.51 | -2.58 | -5.56 | 1.24 | 3.80 | 9.89 | -0.36 | 0.39 | 0.06 | 1.03 | 1.12 | 2.96 |

MIN was up to -93.21 meters in the N direction of HKWS. The corresponding BIAS and RMSE were also large. It is supposed
that this irregular behaviour might be related to the fact that the ionospheric activity at the low latitudes is more intense than
at high latitudes. The corresponding time series of positioning errors during the storm is shown in Fig. 5. The positioning
errors show large fluctuations at the beginning of the storm. That was noticeable in the N and U directions. There were also
minor fluctuations in the E direction. The local time (LT) corresponding to the start epoch of the storm is around 14h, when the
ionospheric activity is the most intense within a day. Thenceforth, there were large jumps in the positioning errors, especially
in the N and U directions. Furthermore, there were no positioning estimations for many epochs in all directions. The strongest
jumps happened at 20 LT, lasting until 8 LT in the next morning. Moreover, there were other jumps around 20 LT in the recovery
phase. These jumps are clearly visible in all the three directions. The number of satellites and PDOP for the three stations are
demonstrated in Fig. 6. It can be seen that there were large jumps in PDOP. The maximum could be at the level of several
hundred. Combining with the number of satellites, these jumps were caused directly due to the loss of tracking satellites. The
similar phenomenon can be found on the next day (DOY 330, 2016) as well. There are many reasons for the failure of tracking
satellites, such as issues on the receiver hardware or software, signal strength, space weather, etc. In this study, the main reason
for the jumps in the positioning errors for the three stations (LHAZ, HKWS, HKSL) may be attributed to a comprehensive



**Table 5.** Statistics indices for BDS B1 frequency positioning errors during moderate storms (units: m)

| MJD | SITEN | MIN | | | MAX | | | BIAS | | | RMSE | | |
|---|---|---|---|---|---|---|---|---|---|---|---|---|---|
| 57181 | gmsd | -1.66 | -2.05 | -10.40 | 0.42 | 1.40 | 2.41 | -0.46 | 0.15 | -3.84 | 0.63 | 0.50 | 5.01 |
| | jfng | -2.26 | -2.90 | -5.98 | 2.23 | 2.29 | 9.07 | -0.67 | -0.15 | 1.09 | 1.09 | 1.15 | 3.82 |
| 57274 | gmsd | -4.13 | -2.02 | -14.74 | 1.47 | 5.17 | 6.92 | -1.42 | 0.83 | -3.69 | 1.76 | 1.71 | 5.40 |
| | jfng | -2.64 | -2.25 | -10.34 | 0.60 | 2.98 | 7.24 | -1.02 | 0.15 | -1.61 | 1.32 | 1.07 | 3.44 |
| 57407 | gmsd | -3.26 | -4.09 | -8.82 | 0.35 | 3.86 | 4.35 | -1.70 | 0.04 | -1.54 | 1.83 | 1.27 | 2.76 |
| | jfng | -2.57 | -2.99 | -5.18 | 0.31 | 2.41 | 5.55 | -1.30 | -0.06 | 0.05 | 1.42 | 0.96 | 2.03 |
| | hkws | -2.11 | -2.66 | -5.93 | 1.59 | 3.89 | 9.51 | -0.69 | 1.13 | 2.28 | 1.16 | 1.86 | 3.86 |
| | hksl | -2.22 | -2.97 | -4.11 | 1.55 | 3.99 | 11.98 | -0.70 | 1.16 | 2.97 | 1.24 | 1.87 | 4.26 |
| 57839 | gmsd | -2.29 | -1.30 | -7.93 | 1.58 | 4.06 | 9.35 | -0.72 | 0.73 | -0.39 | 0.94 | 1.07 | 3.67 |
| | jfng | -1.87 | -1.65 | -4.70 | 2.13 | 2.69 | 11.78 | -0.19 | 0.40 | 1.41 | 0.90 | 0.85 | 4.01 |
| | lhaz | -1.47 | -0.89 | -5.93 | -0.02 | 0.60 | -0.18 | -0.92 | -0.24 | -2.47 | 0.98 | 0.37 | 2.78 |
| | hkws | -1.59 | -2.99 | -9.33 | 0.63 | 2.32 | 3.23 | -0.43 | -0.09 | -1.39 | 0.64 | 1.17 | 3.06 |
| | hksl | -1.61 | -3.16 | -8.81 | 0.33 | 2.47 | 3.93 | -0.56 | -0.25 | -1.22 | 0.73 | 1.22 | 3.05 |
| 58065 | daej | -1.30 | -2.07 | -10.97 | 1.84 | 4.20 | 4.34 | 0.33 | 0.83 | -2.57 | 0.66 | 1.27 | 3.55 |
| | gmsd | -1.48 | -2.00 | -6.87 | 2.01 | 3.13 | 4.18 | 0.27 | 0.34 | -1.28 | 0.61 | 0.84 | 2.33 |
| | jfng | -1.55 | -1.56 | -5.70 | 1.76 | 2.53 | 2.80 | 0.05 | 0.50 | -1.98 | 0.48 | 0.90 | 2.48 |
| | lhaz | -1.68 | -1.26 | -4.74 | 1.71 | 2.16 | 7.08 | -0.18 | 0.17 | -0.78 | 0.70 | 0.65 | 2.20 |
| | hkws | -1.86 | -1.72 | -3.89 | 1.28 | 3.72 | 2.53 | 0.06 | 0.37 | -0.78 | 0.44 | 1.00 | 1.62 |
| | hksl | -1.86 | -1.76 | -3.87 | 1.12 | 3.84 | 2.54 | -0.07 | 0.35 | -0.76 | 0.42 | 1.02 | 1.70 |
| MEAN | | -2.07 | -2.23 | -7.28 | 1.20 | 3.04 | 5.72 | -0.54 | 0.33 | -0.87 | 0.94 | 1.09 | 3.21 |
| MEDIAN | | -1.86 | -2.05 | -5.98 | 1.47 | 2.98 | 4.35 | -0.56 | 0.34 | -1.22 | 0.90 | 1.07 | 3.06 |

effect. The issues on the specific receiver (LEICA version, see Table 3) might be the most possible reason, which caused similar jumps for these three stations. In addition, the time series of related space weather indices are shown in Fig. 7. The vertical

dark grey dotted line indicates the epoch of minimum Dst, the UT time of minimum Dst is labeled in the figure as well. The time range covered 10 days before and after the main phase day. From the figure, the southward IMF Bz had high-frequency variations during the storm period. The solar wind speed increased as well. There were also continual variations in Kp and Dst during that period. These imply that the storm activity became complicated during this period. After removing these three stations, the final statistics are illustrated in Table 7. As shown in the table, the MEAN of RMSEs in the three directions for

B1 positioning errors was 0.92, 1.06, 2.70 meters, and the MEDIAN was 0.82, 0.96, 2.60 meters. The statistics are lower than those for strong and moderate storms.

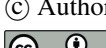



**Table 6.** Statistics indices for BDS B1 frequency positioning errors during weak storms (units: m)

| MJD | SITEN | MIN | | | MAX | | | BIAS | | | RMSE | | |
|---|---|---|---|---|---|---|---|---|---|---|---|---|---|
| 57545 | gmsd | -2.28 | -3.36 | -9.62 | 1.98 | 3.89 | 4.97 | -0.42 | 0.32 | -2.04 | 0.95 | 1.50 | 3.17 |
| | jfng | -2.32 | -3.30 | -6.13 | 0.62 | 4.24 | 5.24 | -0.67 | 0.29 | -0.58 | 0.82 | 1.35 | 2.18 |
| | lhaz | -4.40 | -3.98 | -19.90 | 2.46 | 7.16 | 9.04 | 0.42 | 0.11 | 1.40 | 0.98 | 1.27 | 3.60 |
| | hkws | -2.63 | -4.36 | -7.30 | 0.65 | 4.83 | 9.12 | -0.82 | 0.23 | 1.41 | 0.99 | 1.95 | 3.07 |
| | hksl | -2.89 | -4.64 | -6.25 | 0.78 | 4.87 | 9.27 | -0.88 | 0.23 | 1.56 | 1.10 | 1.95 | 3.00 |
| 57717 | gmsd | -3.89 | -2.87 | -13.36 | -0.09 | 1.96 | 6.45 | -2.00 | -0.23 | -2.67 | 2.15 | 0.86 | 4.87 |
| | jfng | -3.07 | -2.89 | -6.26 | -0.58 | 2.03 | 4.75 | -1.92 | -0.56 | -0.38 | 1.98 | 0.94 | 2.09 |
| | lhaz | -3.01 | 71.06 | -39.24 | 3.07 | 29.86 | 16.22 | -0.99 | -3.19 | 0.11 | 1.43 | 10.71 | 5.59 |
| | hkws | -3.06 | 93.21 | -39.35 | 1.54 | 14.19 | 10.45 | -1.26 | -12.34 | -3.34 | 1.76 | 23.58 | 9.12 |
| | hksl | -2.97 | 84.15 | -28.16 | 1.49 | 50.08 | 23.53 | -1.20 | -13.47 | -3.46 | 1.69 | 30.80 | 11.09 |
| 57785 | gmsd | -1.97 | -1.36 | -6.58 | 0.67 | 2.25 | 2.89 | -0.63 | 0.44 | -2.49 | 0.80 | 0.77 | 3.03 |
| | jfng | -2.06 | -1.55 | -5.41 | 0.08 | 2.13 | 4.08 | -0.81 | 0.33 | -1.32 | 0.89 | 0.66 | 2.03 |
| | lhaz | -3.03 | -2.44 | -8.28 | 0.73 | 1.77 | 7.53 | -0.57 | -0.32 | 1.10 | 0.83 | 0.84 | 2.61 |
| | hkws | -1.67 | -1.20 | -4.29 | 0.20 | 2.28 | 6.27 | -0.68 | 0.67 | -0.19 | 0.79 | 0.92 | 2.45 |
| | hksl | -1.57 | -1.14 | -3.91 | 0.14 | 2.15 | 6.10 | -0.77 | 0.67 | -0.02 | 0.85 | 0.90 | 2.44 |
| 57920 | daej | -2.80 | -2.02 | -8.70 | 1.06 | 3.21 | 3.15 | -0.91 | 0.55 | -2.09 | 1.05 | 1.05 | 3.12 |
| | gmsd | -2.77 | -1.91 | -10.43 | 0.85 | 2.12 | 2.43 | -0.98 | 0.27 | -2.46 | 1.14 | 0.80 | 3.29 |
| | jfng | -2.10 | -1.88 | -3.52 | 0.51 | 2.04 | 3.48 | -0.60 | 0.17 | 0.19 | 0.76 | 0.74 | 1.40 |
| | lhaz | -1.71 | -4.44 | -0.80 | 0.61 | 7.71 | 8.47 | -0.63 | 0.23 | 3.72 | 0.80 | 1.47 | 4.11 |
| | hkws | -2.00 | -3.11 | -1.73 | 0.28 | 2.93 | 4.95 | -0.58 | -0.50 | 0.98 | 0.76 | 1.11 | 1.76 |
| | hksl | -1.96 | -3.21 | -1.81 | 0.53 | 1.73 | 4.89 | -0.44 | -0.54 | 1.00 | 0.73 | 1.07 | 1.74 |
| 58270 | daej | -2.43 | -2.66 | -9.93 | 1.42 | 2.95 | 9.50 | -0.18 | 0.30 | -1.83 | 0.61 | 0.77 | 2.58 |
| | gmsd | -1.72 | -1.99 | -7.34 | 1.28 | 2.06 | 1.10 | -0.17 | 0.11 | -2.75 | 0.52 | 0.60 | 3.15 |
| | jfng | -2.00 | -2.05 | -5.99 | 1.13 | 2.25 | 2.46 | -0.15 | 0.38 | -2.10 | 0.61 | 0.83 | 2.52 |
| | lhaz | -1.96 | -1.51 | -5.99 | 1.56 | 2.03 | 7.14 | -0.46 | 0.70 | -1.44 | 0.82 | 0.97 | 2.71 |
| | hkws | -0.83 | -0.89 | -4.83 | 1.33 | 2.42 | 1.86 | 0.23 | 0.88 | -1.44 | 0.51 | 1.08 | 1.95 |
| | hksl | -0.93 | -0.65 | -4.18 | 1.32 | 2.07 | 1.65 | 0.17 | 0.81 | -1.32 | 0.53 | 0.97 | 1.81 |
| MEAN | | -2.37 | 11.40 | -9.97 | 0.95 | 6.19 | 6.56 | -0.66 | -0.87 | -0.76 | 0.99 | 3.35 | 3.35 |
| MEDIAN | | -2.28 | -2.66 | -6.26 | 0.78 | 2.28 | 5.24 | -0.63 | 0.23 | -1.32 | 0.83 | 0.97 | 2.71 |



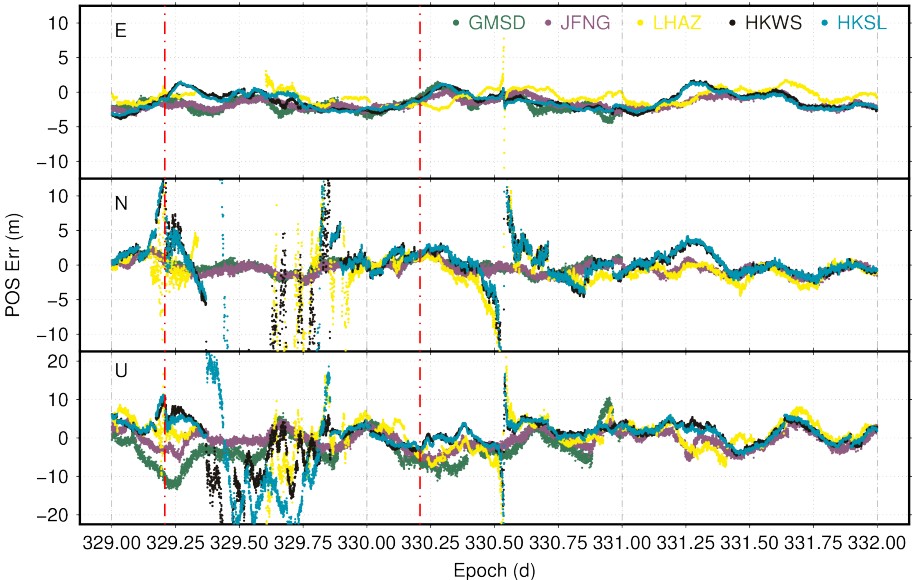

**Figure 5.** Time series of positioning errors for BDS B1 around the main phase around DOY 330, 2016 (X–axis in GPST, Y–axis in meters)

**Table 7.** Statistics indices for BDS B1 positioning errors during weak storms without singular stations (units: m)

| MJD | MIN | | | MAX | | | BIAS | | | RMSE | | |
|---|---|---|---|---|---|---|---|---|---|---|---|---|
| MEAN | -2.29 | -2.48 | -6.77 | 0.81 | 3.05 | 5.28 | -0.60 | 0.23 | -0.57 | 0.92 | 1.06 | 2.70 |
| MEDIAN | -2.08 | -2.25 | -6.19 | 0.70 | 2.25 | 4.96 | -0.62 | 0.28 | -0.95 | 0.82 | 0.96 | 2.60 |

## 4 Conclusions

In this study, the performance of BDS B1 frequency SPP during the main phase of different classes of storms in China and its surrounding area was investigated. From the results, it was observed that the positioning accuracy is affected to different

levels during the storms. Some relevant conclusions can be drawn from the analyses. Firstly, the probability of the extrema in the statistics of positioning errors during strong storms is greatest, followed by moderate and weak storms. Secondly, during the same class of storms the positioning accuracy may vary. Thirdly, the positioning accuracy may be influenced even in the recovery phase of storms.

The findings in this study will contribute to the prediction of BDS positioning accuracy under different strengths of geo-

175 magnetic storms. Besides, the influence of storms could be comparable to other GNSS systems like GPS. Thus, the findings could also be beneficial to those systems. However, since the study period was in the descending phase of solar cycle 24, the influence of space weather events on the ionospheric activity was not intense. Therefore, the effects on the positioning accuracy might not be entirely apparent. The study needs to be extended with the arrival of solar cycle 25 and with the addition of more





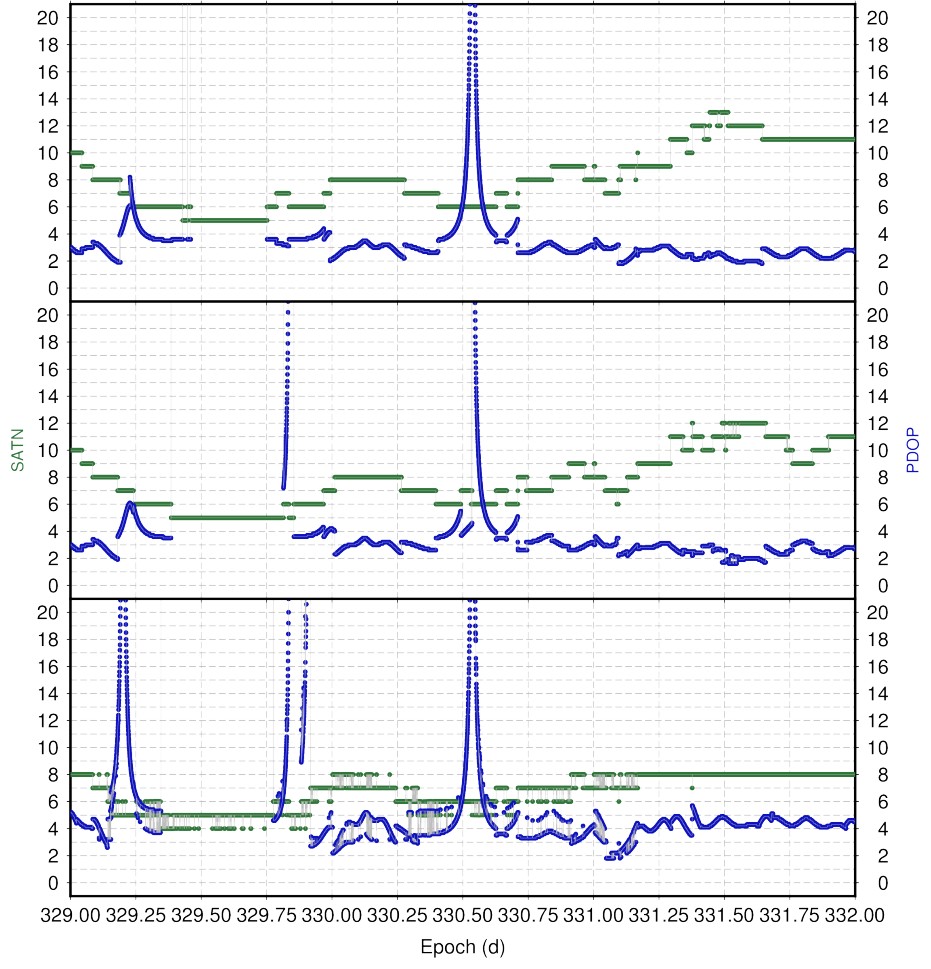

**Figure 6.** The number of satellites and PDOP for station HKSL, HKWS and LHAZ (upper to lower) around DOY 330, 2016 (X–axis in GPST, left Y–axis for the number of satellites in count, right Y–axis for PDOP in count)

storm events. Moreover, with the increase of BDS observation more comprehensive study on the performance of BDS single
frequency SPP can be performed in near future.

*Data availability.* The datasets analysed during the current study are available in the IGS MGEX repository [ftp://cddis.gsfc.nasa.gov/pub] and NASA OMNI [https://omniweb.gsfc.nasa.gov].

*Author contributions.* Junchen Xue, Sreeja Vadakke Veettil and Marcio Aquino designed the experiments. Junchen Xue carried out analysis and prepared the paper. All co-authors contributed to the reading, interpretation and comments of the paper.

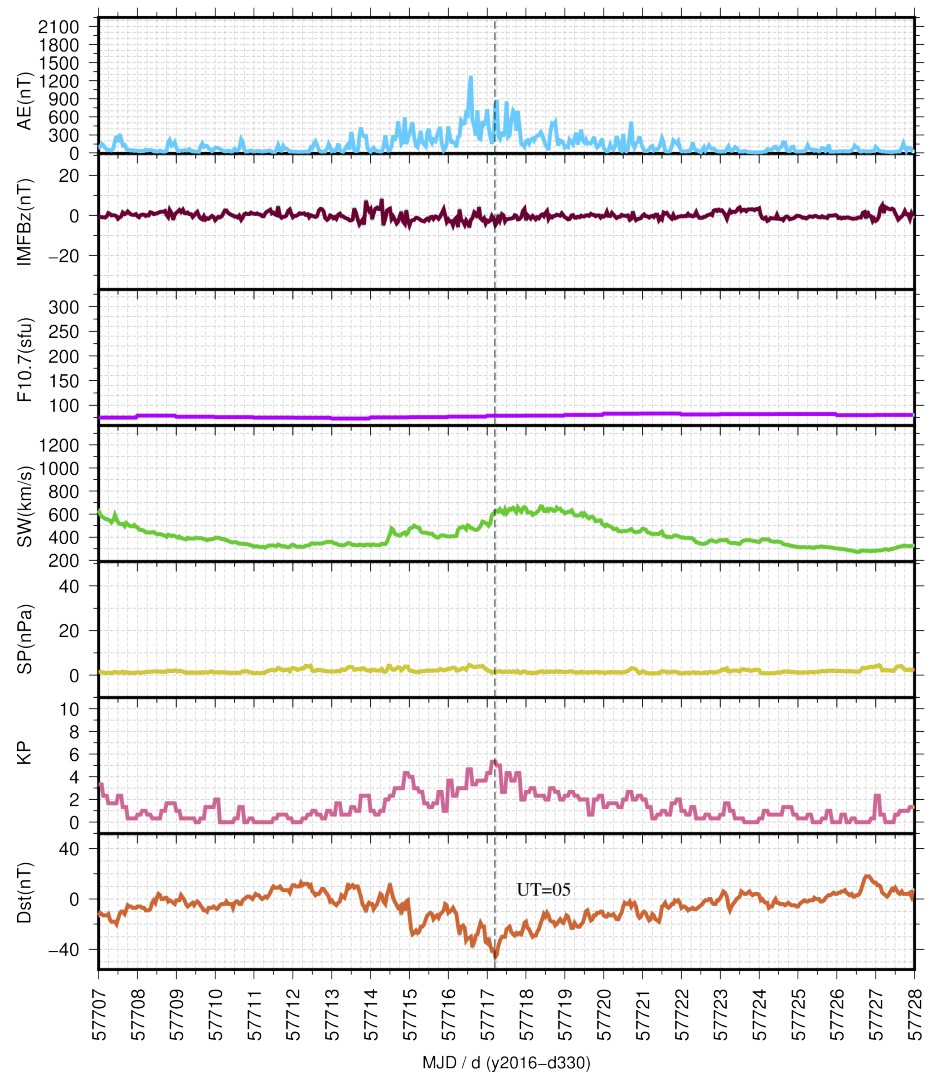

**Figure 7.** Time series of space weather indices around the main phase on DOY 330, 2016(X–axis in UT)

*Competing interests.* The authors declare that they have no conflict of interest.

*Acknowledgements.* The authors thank the anonymous referees for their valuable suggestions. This study is supported by National Natural Science Foundation of China (Grant No. 11703066) and a scholarship from the China Scholarship Council (CSC).



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
