# Peer review of "Performance of BDS B1 frequency standard point positioning during the main phase of different classes of geomagnetic storms in China and its surrounding area"

_Annales Geophysicae, 2021_

## Referee Comment (RC2)

[revised manuscript text omitted]

---

## Author Comment (AC1)

Ann. Geophys. Discuss., referee comment RC1
https://doi.org/10.5194/angeo-2021-5-RC1, 2021

[Figure]

**Comment on angeo-2021-5**

Anonymous Referee #1
* * *
Referee comment on "Performance of BDS B1 frequency standard point positioning during the main phase of different classes of geomagnetic storms in China and its surrounding area" by Junchen Xue et al., Ann. Geophys. Discuss., https://doi.org/10.5194/angeo-2021-5-RC1, 2021
* * *
The paper analyzes the BeiDou B1 frequency standard point positioning in China and its surrounding area during selected magnetic storm events from 2015 to 2018 and they pointed out that positioning accuracy was deteriorated during the storm. The positioning error was larger for stronger magnetic storms. The root mean square error (RMSE) in position for the different magnetic storms in the East, North and Up directions were also presented. This topic has been discussed previously in the literature and the original contribution of this paper is the fact that the data were from Beidou B1 frequency. Some improvements and clarifications need to be done before the paper could be accepted to Annales Geophysicae. Please see the below comments.

**– The authors just analyzed the Bias and the bias RMSE (Figures 1-3 and Tables 4 â" 6). They did not provide the precision of the positioning, that come from the Covariance matrix;**
**- Would be quite important also to show the precision from the adjustment, as well as the quality control of the adjustment;**
**- Why the standard deviation was not shown? The coordinate obtained has an uncertainty, which in some cases may even be greater than discrepancy. In this case, it appears that the uncertainty (standard deviation) was considered to be zero or disregarded. Not being zero, the standard deviation impacts in the coordinate accuracy;**
A: As indicated by the reviewer, the standard deviation (STD) is quite important to show the quality control of the adjustment. The STD can be shown but it is only the reference for the application. The comparison of estimations with the true ground coordinates from IGS or ITRF is the final and accurate validation, which provides the actual positioning accuracy. That's why most of the literature show the external accuracy validation with IGS or ITRF products. The indices for validation are BIAS and RMSE or MAE.
We have computed the standard deviation from the adjustment and exemplified them below. But how to show them in a table is a big issue. It seems that they should not be put together with indices like BIAS, RMSE, etc for the external validation.

[Figure]

Time series of positioning error STDs for BDS B1 frequency during a strong storm around DOY 238, 2018 (X–axis in GPST, Y–axis in meters)

[Figure]

Time series of positioning error STDs for BDS B1 frequency during a moderate storm around DOY 086, 2017 (X–axis in GPST, Y–axis in meters)

[Figure]

Time series of positioning errors STDs for BDS B1 frequency during a weak storm around DOY 032, 2017 (X–axis in GPST, Y–axis in meters)

**What is the Klobuchar model contribution to the positioning error since it corrects about 40 to 50% of the ionospheric effect? A discussion or even some quantitative values should be presented in the paper;**
A: The mean correction precision of the Klobuchar-style ionospheric navigation model used in BDS is better than 65%. That is better in middle latitudes than in low latitudes. The details can be found in (Wu, Hu et al. 2013)( Wu, X., et al. (2013). "Evaluation of COMPASS ionospheric model in GNSS positioning." Advances in Space Research 51(6): 959-968.). The related discussion has now been inserted in the 'Introduction' part.

**- Figure 2: there is no data for LHAZ between day 86 and day 87 during the moderate storm. Mention this fact and explain the reason of this lack of data;**
A: This fact was already mentioned in the manuscript. But the reason for the missing data is unavailable as there are no descriptions regarding this in the IGS data centers.

**- Figure 4: I would expect it to be explored in the paper;**
A: please see lines 110-117 in the modified manuscript.

**- Figures 1 to 3: plot in the top the simultaneous DST or even better, if available, the SYM H (instead of Dst) that has a time resolution of 1 minute;**
A: The Dst has been appended to the top of each figure.

**- Explain in details what could be the ionospheric activity at low latitudes mentioned at line 145 and include the explanation in the paper;**
A: As a comparison, there were no big changes in the positioning errors for other stations (GMSD, JFNG) which lie in the higher latitude. So the sentence here is a reasonable guess for the issue. The related sentence has been modified in the manuscript.

**- The title should include recovery phase since results from this storm phase are also presented. As a suggestion, even though it is too large (try to shorten it) : âœPerformance of BDS B1 frequency standard point positioning during the main and recovery phases of different classes of geomagnetic storms in China and its surrounding areaâ□□;**

A: The main part in this manuscript discusses the statistics of positioning errors during the main phase of the storm. In our opinion, the title of this manuscript should not be changed.

**- Are there severe storms according to Astafyeva et al., 2014 classification (Dst>> caused 019 include here the Aarons paper (see reference below) * 104 check if there is Solar Radio Burst (SRB) around 01 LT since SRBs can cause positioning errors**
A: There were no severe storms during that time. Please see the Dst values in Table 2. We have checked the SRB events from the NOAA database. There was no SRB event around 01 LT. In addition, if there was a SRB around 01 LT, the stations in the night-side of hemisphere will not be disturbed.

**119 What is the effect of different versions of the receiver hardware on the positioning calculations?**
A: The noise of observations depends on the version of the receiver hardware. In some challenging situations like ionospheric scintillation, receivers with different hardware versions will experience different effects like on signal tracking.

**125 Table 4 â" Table 6 (just missing one space) 131 1 â" 4 m (just missing one space) 138 range of 1 â"3 m (just missing one space)**
A: As suggested by the reviewer, the tables have been corrected and updated in the revised manuscript.

**148 same of line 104: check the possibility of Solar Radio Burst (SRB) occurrence 163 complicated >>> complex * Aarons J (1991) The role of the ring current in the generation or inhibition of equatorial F-layer irregularities during magnetic storms. Radio Sci 26:1131â"1149**
A: The indices events were checked around DOY 2016/330 but no SRB events which can affect the positioning of GNSS signals were found.

Downloaded by [794000] researchgate.org

---

## Author Comment (AC2)

Many thanks to the reviewer. Some comments are contributive to this manuscript. The following illustrates answers to the questions by the reviewer.

A. Questions in the general comment:

Q1: First, the motivation of the paper is not clear. Does it target single-frequency users of GNSS and try to give lessons for the future use of the system? Does it aim to compare BDS B1 frequency results with the L1 SPP frequency of the GPS? Or, does it aim to compare the findings of the study with those of the BDS studies which were previously published? Neither a literature review nor comparisons of results to previous studies are provided relating to the above questions. the motivation of the study is not clearly stated in the abstract ad in the introduction

A: The findings in this study could contribute to the prediction of BDS positioning accuracy under different classified geomagnetic storms, and it could be beneficial to other systems such as BDS-3 as well. The motivation is now explicitly mentioned in the Abstract. Actually, the motivation was previously mentioned in the Conclusions.

The topic and results presented in this manuscript are novel and new, therefore, there was no existing literature exactly related to the topic. All the existing literature corresponds to the cause of the geomagnetic storms and the effects of geomagnetic storms on the GNSS applications. The state of the art literature review related to the topic of this manuscript is mentioned in the introduction.

Q2: the sampling strategy is not discussed well in the beginning and the weaknesses related to those are stated in the conclusion! Was that possible to adopt a better sampling strategy from the rich IGS network!?

A: The strategy was explained in Section 2, but as suggested one more sentence about the sampling strategy of the data is now added in Section 2. It is to be noted that this manuscript aims to study the effects of geomagnetic storms on the BDS application in China and its surrounding area. Therefore, only MGEX network for China area was applied to get the BDS observation. The earliest period to get the BDS observation is from 2015. In addition, the traditional IGS network only provided GPS and GLONASS observations before they moved data from MGEX to IGS.

Q3: the authors determined that some days with strong storms do not affect positioning accuracy but the authors do not refer to literature and include discussion for possible underlying facts. These are serious weaknesses of the paper and need to be improved for the next submission.

A: This was just the important finding in this manuscript, it is not necessary consequence that strong storm must have effects on the positioning accuracy. The possible reasons for this finding is discussed in the manuscript. But it should be noted that the causes during geomagnetic storms could be quite complicated and this is not the aim of this manuscript. The main aim of this manuscript is to study the effects of the main phase of different classified geomagnetic storms on the performance of BDS B1 SPP in China area. The manuscript aims to give statistical results of those effects, and reach valuable conclusions for the related studies

and further GNSS applications. As suggested, a relevant literature is now added in the manuscript.

**B. Questions in attachment:**

**Q1: What is the real motivation of this study?**
**A:** The answer is in the answer to A.Q1.

**Q2: What is meant by this sentence?**
**A:** This means that the positioning accuracy during similar classes of storms (for example strong storms) need not be at the same level and that there is no positive linear correlation between storm level and positioning accuracy.

**Q3: what is the accuracy of kinematic positioning anyway?**
**A:** the kinematic positioning accuracy was degraded during storms, the repeatability of it reached 12.8, 8.1 and 26.1cm in ENU directions.

**Q4: What are those papers?**
**A:** the answer is in the answer to A.Q1. The study in this manuscript was performed for the first time and there is no existing literature. The grammar here means negative.

**Q5: not clear**
**A:** comparing with other positioning modes, SPP is more obviously affected by ionospheric delays.

**Q6: why solar cycle 24?**
**A:** Because the earliest period for collecting BDS data is in solar cycle 24. We cannot get BDS observations in the China area for other cycles.

**Q7: why this period selected?**
**A:** the answer is the same as to B.Q6!

**Q8. What software was used?**
**A:** we developed our software with the fundamentals of SPP.

**Q9: station coordinates in Cartesian coordinate system do not relate to the coordinates ENU?**
**A:** Right! we showed the processing in the whole paragraph. Coordinates from SPP was compared with those in the SINEX files. The conversion to local site coordinate frame was then performed and the positioning errors are finally given in the three directions. The statistics was made based on the positioning errors. We thought the processing is well known in the GNSS community. But thanks anyway. Some more information is now illustrated in this context.

**Q10: this bit here not clear!?**

**A:** The paragraph is organized well and presented clearly. The paragraph illustrates the solutions of BDS B1 SPP during different classes of storms and give basic descriptions to the figures.

**Q11: why not the storm fall into the main phase? &**

**why does the disturbance caused by the storm not fall into the main phase given with vertical read dashed lines?**

**A:** This can be attributed to the fact that the Dst values are lower and the geomagnetic disturbance remained intense throughout.

**Q12: LT local time?**

**A:** Yes, LT is local time. The term was used in the Introduction part, please see line 18 in the original manuscript.

**Q13: I don't think the GPS community will understand the lines 104 through 107! Add some more explanation (i.e. related to space weather indices)**

**A:** The explanation was made combining the space weather indices such as F10.7cm radio flux. The values for the F10.7cm radio flux were checked by space weather indices.

**Q14: did you also use it for the analysis?**

A: The results by BNC were used in the analysis, please see Figure 4 and 6.

**Q15: how?**

**A:** The conclusion can be found from Table 4—7.

**Q16: MJD is not appropriate to compare these!!**

**A:** Only one MJD term was used in the text and also indicated by Year and DOY. See line 142-143 in the original manuscript. It is clear. The corresponding Year and DOY to MJD was also shown in Table 2.

**Q17: Then what!?**

**A:** The sentence is clear and all right.

**Q18: comparison to GPS literature would be useful to assess the above given figures. & how about comparison to available studies? Both GPS and BDS? (in line 180)**

**A:** That seems a good idea. But as expressed in the last two sentences in A.Q3, the idea might be the further study.

**Q19: the reader is not able to associate the large deviatioins with doys having storms unless MJD with high storms are denoted before! &**

**why not use GPS doys instead?**

**A:** Using MJD is right for the table shown, the related Year and DOY to each MJD are also

shown in Table 2.

**Q20: why? Reference.**
**A:** A reference was added as suggested.

**Q21: did you check Kp index for the above given GPS day?**
**A:** Yes, there are similar phenomena for Kp indices during those days. But we used Dst instead of Kp owing to the much better time resolution and accuracy of Dst to indicate geomagnetic storms.

**Q22: describe space weather indices prior to this paragraph!**
**A:** We think the location of description of space weather indices is all right and logical.

**Q23: why? Refer to literature!**
**A:** The conclusions were made based on all materials (the figures and tables) and discussions in the manuscript. No literature is needed.

**Q24: this could even be done here!**
**A:** The answer is the same with that in B.Q18.

**Q25: then why not a proper sampling scheme adopted?**
**A:** We did not have BDS data in China area for solar cycle 25 during the preparation of this manuscript. The solar cycle 25 is just beginning! Please also find the answer in A.Q2 and B.Q6.

**Q26: kp indiex is more frequently referred by the GPS community to spot days with high geomagnetic activity.**
**A:** Kp is the mean of K index, which is not enough to indicate geomagnetic activities. Actually, in space physics community, Dst or SYM-H is the right one for the study of geomagnetic activities. We recommend Dst or SYM-H can be used in GNSS community in future study. Actually, there are some uses of Dst or SYM-H by GNSS scholars (see references herein).